# Heritability of Urinary Amines, Organic Acids, and Steroid Hormones in Children

**DOI:** 10.3390/metabo12060474

**Published:** 2022-05-24

**Authors:** Fiona A. Hagenbeek, Jenny van Dongen, René Pool, Amy C. Harms, Peter J. Roetman, Vassilios Fanos, Britt J. van Keulen, Brian R. Walker, Naama Karu, Hilleke E. Hulshoff Pol, Joost Rotteveel, Martijn J. J. Finken, Robert R. J. M. Vermeiren, Cornelis Kluft, Meike Bartels, Thomas Hankemeier, Dorret I. Boomsma

**Affiliations:** 1Department of Biological Psychology, Vrije Universiteit Amsterdam, 1081 BT Amsterdam, The Netherlands; j.van.dongen@vu.nl (J.v.D.); r.pool@vu.nl (R.P.); m.bartels@vu.nl (M.B.); di.boomsma@vu.nl (D.I.B.); 2Amsterdam Public Health Research Institute, 1081 BT Amsterdam, The Netherlands; 3Amsterdam Reproduction & Development (AR&D) Research Institute, 1081 BT Amsterdam, The Netherlands; 4Division of Analytical Biosciences, Leiden Academic Center for Drug Research, Leiden University, 2333 AL Leiden, The Netherlands; a.c.harms@lacdr.leidenuniv.nl (A.C.H.); n.karu@lacdr.leidenuniv.nl (N.K.); hankemeier@lacdr.leidenuniv.nl (T.H.); 5LUMC-Curium, Department of Child and Adolescent Psychiatry, Leiden University Medical Center, 2342 AK Oegstgeest, The Netherlands; peter.roetman@gmail.com (P.J.R.); r.r.j.m.vermeiren@lumc.nl (R.R.J.M.V.); 6Neonatal Intensive Care Unit, Department of Surgical Sciences, University of Cagliari, 09124 Cagliari, CA, Italy; vafanos@tiscali.it; 7Department of Pediatric Endocrinology, Emma Children’s Hospital, Amsterdam UMC, location VUmc, 1081 HV Amsterdam, The Netherlands; b.j.vankeulen@amsterdamumc.nl (B.J.v.K.); j.rotteveel@amsterdamumc.nl (J.R.); m.finken@amsterdamumc.nl (M.J.J.F.); 8Centre for Cardiovascular Science, Queen’s Medical Research Institute, University of Edinburgh, Edinburgh EH16 4TJ, UK; b.r.walker@ed.ac.uk; 9Institute of Genetic Medicine, Newcastle University, Newcastle upon Tyne NE1 7RU, UK; 10Department of Experimental Psychology, Helmholtz Institute, Utrecht University, 3584 CS Utrecht, The Netherlands; h.e.hulshoff@umcutrecht.nl; 11Department of Psychiatry, Brain Center Rudolf Magnus, University Medical Center Utrecht, 3584 CX Utrecht, The Netherlands; 12Youz, Parnassia Psychiatric Institute, 2566 ER The Hague, The Netherlands; 13Good Biomarker Sciences, 2333 CL Leiden, The Netherlands; kluft@kluft.in

**Keywords:** heritability, classical twin design, metabolites, urine, children, amines, organic acids, steroid hormones

## Abstract

Variation in metabolite levels reflects individual differences in genetic and environmental factors. Here, we investigated the role of these factors in urinary metabolomics data in children. We examined the effects of sex and age on 86 metabolites, as measured on three metabolomics platforms that target amines, organic acids, and steroid hormones. Next, we estimated their heritability in a twin cohort of 1300 twins (age range: 5.7–12.9 years). We observed associations between age and 50 metabolites and between sex and 21 metabolites. The monozygotic (MZ) and dizygotic (DZ) correlations for the urinary metabolites indicated a role for non-additive genetic factors for 50 amines, 13 organic acids, and 6 steroids. The average broad-sense heritability for these amines, organic acids, and steroids was 0.49 (range: 0.25–0.64), 0.50 (range: 0.33–0.62), and 0.64 (range: 0.43–0.81), respectively. For 6 amines, 7 organic acids, and 4 steroids the twin correlations indicated a role for shared environmental factors and the average narrow-sense heritability was 0.50 (range: 0.37–0.68), 0.50 (range; 0.23–0.61), and 0.47 (range: 0.32–0.70) for these amines, organic acids, and steroids. We conclude that urinary metabolites in children have substantial heritability, with similar estimates for amines and organic acids, and higher estimates for steroid hormones.

## 1. Introduction

Metabolites are small, low molecular weight (<1.5 kDa) molecules that are involved in cellular metabolism [1]. Metabolites have many functions in the human body, including signaling, energy storage, and structure formation [2]. The extensive profiling of metabolites in biofluids, cells, or organisms, i.e., metabolomics, provides the means to investigate metabolic regulation [3], and has been successfully applied to elucidate biological mechanisms and to identify drug targets or disease biomarkers [4]. Individual differences in metabolite levels reflect variation in exogenous factors, such as diet, medication use, or smoking behavior, and endogenous factors, such as sex, and age [5]. Genetics is an endogenous factor that has both direct and indirect influences through lifestyle and behavior.

Previous heritability studies of metabolomics data in various biofluids found that about 50% of the variance in metabolite levels is due to additive genetic factors [6,7]. Non-additive (i.e., dominant genetic effects) and common (shared) environmental factors (i.e., shared by family members) generally have a minor influence on metabolite levels [8,9,10,11,12,13]. Some nuance on the contribution of genetic factors is possible: the contribution to individual differences in metabolite levels differs per metabolite class and subclass, with generally somewhat higher heritability estimates observed for lipids and lipid-like molecules than for organic acids and derivatives [14,15,16,17,18,19,20,21]. The genetic architecture of many subclasses of metabolites has yet to be studied in detail. For example, steroid hormones, compared with other lipids, are generally not well represented on broad metabolomics platforms, and are less frequently investigated with targeted analytical platforms. Targeted metabolomics studies that include steroid hormones show a substantial contribution of genetic factors to cortisol and cortisol metabolism, androgens, and estrogens in various biofluids [22,23,24,25] and sometimes report sex differences in the contribution of genetic or shared environmental factors to steroid hormones [26].

With the exception of steroid hormones, the earlier studies have mainly focused on metabolite levels in serum or plasma samples in (young) adults of European ancestry. Relatively little is known about the metabolome in children, in different tissues, or in different ancestries. Here, we aimed to elucidate the genetic etiology of the urinary metabolome in children. Urine is an interesting biological matrix for metabolomics studies as it can be collected non-invasively in large quantities. Urine metabolites provide a snapshot of an individual’s exposure and endogenous (water-soluble) waste clearance and reflect filtration of blood in the kidneys at approximately 120mL per minute [27]. Urine contains a wealth of information on metabolic activity, and to date more than 3000 metabolites have been identified in urine [28,29]. One investigation of the urinary metabolome in children reported high estimates for SNP-based heritability [30], suggesting a substantial contribution of genetic factors to urinary metabolite levels in children.

The ACTION Biomarker Study was designed to investigate the etiology of childhood aggression and to assess the role of the genome, epigenome, and metabolome in explaining differences between children in aggressive behaviors [31,32,33]. To assess the metabolome, a targeted approach was chosen to cover metabolite classes previously associated with aggressive behavior [34], and urine samples were collected in twins from the Netherlands Twin Register [35] (NTR-ACTION cohort) and in an independent clinical cohort (LUMC-Curium cohort). Specifically, urinary metabolomics data for 86 metabolites were generated on a liquid chromatography mass spectrometry (LC-MS) platform targeting amines, a gas chromatography (GC) MS platform targeting organic acids, and an LC-MS platform targeting steroid hormones. Here, we first investigate the effects of sex and age on these 86 urinary metabolites in the NTR-ACTION cohort comprising 1068 monozygotic (MZ) twins (531 pairs, *M* age = 9.6, *SD* age = 1.9, 47.4% females) and 232 dizygotic (DZ) twins (114 pairs, *M* age = 9.9, *SD* age = 1.6, 51.7% females, Table 1). We undertook to replicate the sex and ages associations in the LUMC-Curium cohort of 179 children (*M* age = 10.2, *SD* age = 1.8, age range: 6.3–13.4, 25.1% females, Table 1). Second, we aim to characterize the reliability of our measures by considering the MZ twin correlations. The MZ correlation forms an alternative for test–retest correlation, and, as, such, can serve as a reliability index. Specifically, a high MZ correlation is consistent with high reliability of the measure. However, a low MZ correlation does not necessarily mean that the reliability is low [36]. Third, we quantified the familial resemblance for urinary metabolites in children by estimating the correlations in MZ and DZ twin pairs and applied genetic structural equation modeling of MZ and DZ data to obtain maximum likelihood estimates of heritability. We found that sex and age significantly influence urinary metabolites in young twins and replicated the associations with age in the independent clinical cohort. The MZ twin correlations implied that we could measure approximately 90% of the urinary metabolites with moderate to high reliably, with more steroids rating high reliability as compared to amines and organic acids. Genetic modelling revealed that genetic factors account for nearly all familial resemblance in urinary metabolites.

## 2. Results

### 2.1. Urinary Metabolites in Children Have Robust Associations with Age Not with Sex

To explore the effects of sex and age on urinary metabolites, we ran mixed-effect models with sex and age as fixed effects and family as a random effect in 1300 participants (*M* age = 9.6 years, *SD* age = 1.8, age range: 5.7–12.9, 48.2% females) of the NTR-ACTION cohort (Table 1). Sex and age effects were significant for 13 of the 56 amines (23%), for 2 of the 20 organic acids (10%), and for 4 of the 10 steroids (40%) (Figure 1, Appendix A). Age effects, but not sex effects, were significant for 39 (70%), 14 (70%), and 5 (50%) amines, organic acids, and steroid hormones, respectively. For one organic acid (5%) and one steroid (10%), sex effects, but no age effects, were significant. For the remaining 4 amines (7%) and 3 organic acids (15%), we observed no significant effects of sex or age. We replicated the significant associations of age with 30 metabolites in 179 children of the LUMC-Curium cohort (*M* age = 10.2 years, *SD* age = 1.8, age range: 6.3–13.4, 25.1% females, Figure 1, Appendix A). The remaining associations for age, and none of the associations for sex were replicated. There were high correlations among the strength and direction of association between the discovery and replication results (Appendix A).

We expect that certain metabolites, especially steroid hormones, have lower levels in younger children and thus may not show sex differences at younger ages. Therefore, we performed a series of sensitivity analyses by stratifying the sample by sex and into two age groups, consisting of children aged 10 years or older and children younger than 10 years (see Appendix A). The association of sex in the older age group (age ≥ 10) with putrescine showed higher levels in girls, but all other associations with sex or age did not replicate across the sensitivity analyses.

### 2.2. Ninety Percent of Urinary Metabolites in Children Have Moderate to High Reliability

To estimate the lower bound of reliability of the urinary metabolites in children, as obtained with the metabolomics platforms for the ACTION Biomarker Study, we obtained MZ twin correlations for 1068 MZ twins (531 pairs, *M* age = 9.6, *SD* age = 1.9, 47.4% females) (Table 1). The MZ correlations, corrected for sex and age effects (Appendix A), were on average 0.51 for the amines (range: 0.25–0.75), 0.52 (range: 0.33–0.64) for the organic acids, and 0.61 (range: 0.43–0.81) for the steroids (Figure 2, Appendix A). MZ correlations may be interpreted as a form of test–retest reliability [36], and these results thus suggest that approximately 90% of the measures of urinary metabolites in the ACTION Biomarker Study were measured with adequate reliability. In total, 1 (1.8%) amine and 4 (40.0%) steroids were obtained with excellent reliability (*r*_MZ_ ≥ 0.75), 8 (14.3%) amines and 3 (15.0%) organic acids had good reliability (*r*_MZ_ 0.60–0.74), and 43 (76.8%) amines, 13 (65.0%) organic acids, and 6 (60.0%) steroids had moderate reliability (*r*_MZ_ 0.40–0.59) [37]. For the remaining 4 amines (7.1%) and 4 organic acids (20%, *r*_MZ_ < 0.40), we could not infer their reliability, as low MZ correlations do not necessarily imply low reliability [36].

### 2.3. Genetic Inluences Explain Familial Resemblance in Urinary Metabolites in Children

MZ twin correlations reflect both the lower bound of measurement reliability, and the upper limit of the broad-sense heritability (h^2^) of these metabolites, where broad-sense heritability refers to the variance explained by all, i.e., additive and non-additive genetic, factors. When correlations in MZ twins are higher than in DZ twins, this is consistent with a genetic influence on a trait. The more detailed pattern of MZ and DZ correlations is informative to assess the contribution of additive and non-additive genetic factors and of shared environmental contributions to their resemblance.

We obtained MZ and DZ twin correlation for 1068 MZ twins (531 pairs, *M* age = 9.6, *SD* age = 1.9, 47.4% females) and 232 DZ twins (114 pairs, *M* age = 9.9, *SD* age = 1.6, 51.7% females) from the NTR-ACTION cohort (Table 1). Figure 2 clearly shows that for nearly all metabolites, the MZ correlations are substantially larger than the DZ correlations, indicating that genetic factors are an important source of phenotypic individual differences. The mean MZ correlation was 0.51 for the amines (range: 0.25–0.75), 0.52 (range: 0.33–0.64) for the organic acids, and 0.61 (range: 0.43–0.81) for the steroids (Figure 2, Appendix A). The mean DZ correlations were 0.16 (range: 0.01–0.46), 0.23 (range: 0.07–0.35), and 0.25 (range: 0.11–0.44), for the amines, organic acids, and steroids, respectively. Thus, the MZ correlations indicated substantial broad-sense heritability (h^2^ > 0.40) for 52 (92.9%) amines, 16 (80.0%) organic acids, and 10 (100%) steroids.

Based on the twin correlations, we estimated the contribution of genetic and environmental contributions to the phenotypic variance by genetic structural equation modelling, by estimating additive genetic effects (A), unshared environmental effects (E), and dominance effects (D) or common (shared) environmental effects (C). Because the ACDE model is not identified in the classical twin design, we applied a common rule of thumb for inspecting twin correlations [38] and fitted the ADE model given *r_MZ_* > 2×*r_DZ_*, or the ACE model given *r_MZ_* < 2×*r_DZ_* (Figure 2, Appendix A). We estimated variance components in ADE models for 50 (89.3%) amines, 13 (65.0%) organic acids, and 6 (60.0%) steroids, and ACE models for 6 (10.7%) amines, 7 (35.0%) organic acids, and 4 (40.0%) steroids (Appendix A).

For the 69 (80.2%) metabolites where, the ADE model was the model of choice, we reported the broad-sense h^2^, not the separate estimates of the standardized variance for A and D. Given the relatively small number of complete DZ pairs (114) compared with the number of MZ pairs (645) (Table 1), the power to detect these variance components is low [39,40]. Here, the broad-sense h^2^ was 0.49 (range: 0.25–0.64), 0.50 (range: 0.33–0.62), and 0.64 (range: 0.43–0.81) for the amines, organic acids, and steroids, respectively (Figure 3, Appendix A). In the ADE models, the mean contribution of unique environmental factors was 0.51 (range: 0.36–0.75) for the amines, 0.50 (range: 0.38–0.67) for the organic acids, and 0.36 (range: 0.19–0.57) for the steroids. For the 17 (19.8%) metabolites where an ACE model fitted best, the narrow-sense h^2^ (i.e., the standardized effect of A in the ACE model) was 0.50 (range: 0.37–0.68) for the amines, 0.50 (0.23–0.61) for the organic acids, and 0.47 (range: 0.32–0.70) for the steroids (Figure 3, Appendix A). Here, common environmental factors explained on average 13% (range: 0.04–0.28), 6% (range: 0–0.13), and 10% (range: 0.02–0.18) of the variance, whereas unique environmental factors accounted for approximately 37% (range: 0.25–0.46), 45% (range: 0.36–0.64), and 42% (range: 0.21–0.53) of the variance in amines, organic acids, and steroids, respectively.

## 3. Discussion

We investigated the influence of sex and age on 86 urinary metabolites and estimated the heritability of these metabolites in 5- to 13-year-old twins. To assess the association of age and sex we tested their significance in a discovery cohort of twins. Even in this sample with a relatively small age range, age was significantly associated with 50 metabolites (58%) and sex with 21 metabolites (35%). Replication in an independent cohort showed that the associations with age were the most robust, as 30 metabolites retained their significant association with age, whereas the association of putrescine with sex only replicated in the older age group (age ≥ 10) after stratification. To address the question if metabolites can be reliably assessed in urine samples collected at home from young children, we determined the lower bound of the reliability, by estimating the MZ correlations and observed a range of 0.25–0.81, with 90% of the urinary metabolites reliably measured. We addressed the etiology of individual differences by the application of the classical twin design and genetic structural equation modeling of the MZ and DZ data. The main result is that individual differences in urinary metabolites are mainly explained by genetic factors. In 69 (80.2%) metabolites the broad-sense heritability was estimated at approximately 51% (range: 25–81%) and in 17 metabolites the narrow-sense heritability was 49% on average (range: 23–70%), with 9% (range: 0.3–28%) of the variance attributable to common environmental factors.

We reported robust associations of urinary metabolites with age but hardly with sex. We had anticipated sex differences in steroid hormones [26,41], and it might be that these will manifest as the children become older. We observed the most robust associations for age with dehydroepiandrosterone sulfate (DHEA-S), conjugated etiocholanolone (sulfate and glucuronide), and homovanillic acid, as these associations were observed in the discovery and replication cohorts regardless of dichotomization of the age variable (Appendix A). Age affects the activity of endocrine systems, including the hypothalamic-pituitary-adrenal (HPA) axis and the hypothalamic-pituitary-gonadal (HPG) axis. Four conjugates of gonadal androgens were positively associated with age (Figure 4), which reflects expected physiological changes with age. Conversely, four adrenal-produced glucocorticoids were negatively associated with age of children in both cohorts (Figure 4). Higher levels of circulating cortisol are not unexpected in younger children (aged 39–106 months) [42]. High secretion of the corticosteroids is associated with delayed pubertal development [43,44], but is also affected by exposure factors, such as high-protein diet [45]. Therefore, the observed associations of steroidal hormones with age may reflect both the direct impact of age on the secretion of these metabolites, and indirectly reflect age effects on exposure-related factors that change the secretion, metabolism, and urine excretion rates. Additional associations with age were exclusively negative (Figure 1), and many of these metabolites are expected to change along the course of life [46,47,48,49]. They included energy production intermediates and neuroactive compounds (along with their metabolites or precursors), as well as products of lipid and protein metabolism which are routinely measured in urine [49].

In contrast to our findings, a previous study found no associations of age with 44 urinary metabolites, and only three significant associations with sex in a meta-analysis of 1192 children [50]. Part of the discrepancies between our study and the meta-analysis is due to the low overlap in urinary metabolites. Only 16 of the 44 urinary metabolites (36%) included in the meta-analysis were also included on the metabolomics platforms comprising the ACTION Biomarker Study. Of the 16 overlapping metabolites 14 had significant associations with age and 2 with sex, in the discovery cohort of the current study. Another likely explanation for the discrepancy is that the children in the meta-analysis were younger (mean = 7.4, range: 6.5–8.9) than those included in our discovery (mean = 9.6, range: 5.7–12.9) and replication cohorts (mean = 10.2, range: 6.3–13.4). Thus, the metabolites that are significantly associated with age in our study might reflect the onset of puberty. This would explain why the associations with the steroids, DHEA-S and conjugated etiocholanolone were among the most robust.

Genetic modelling showed that all metabolites, irrespective of platform, are subject to genetic influences. We might have expected that in this sample of young children (who live at home with their parents, and share diet, exposures to the same household, and neighborhood), there would be a detectable influence of the shared environment. However, we found that mainly genetic and unique environmental factors accounted for the individual differences in urinary metabolites in children. The substantial contribution of genetic factors to urinary metabolites in children is in line with previously reported SNP-based heritability estimates for the urinary metabolome in children [30]. A previous small twin study (*N* = 128) investigating the urinary metabolome in adults showed a moderate contribution of genetic factors to 14 diet-associated urinary metabolites, and substantial contributions of shared environmental influences [51]. This study in adults only investigated the heritability of 14 diet-associated metabolites, already implying these metabolites are under environmental influence. In this same sample of twins, it was shown that 20% of the urinary NMR metabolomic profile was stable over a 2-month period and that both common genetic and shared environmental factors contribute to the short-term stability of the urinary metabolome in adults [52]. Previous studies investigating the blood metabolome in adult populations [8,10,11,12,13] also obtained very limited evidence for contributions of the common environment, whereas the influence of genetics is substantial. To date, genome-wide association studies for blood metabolites have identified more than 800 metabolite loci, with an average heritability of metabolite loci of 6% for lipids and lipid-like molecules and of 1% for organic acids and derivatives [18]. Here, we extend these findings to show substantial broad-sense heritability for the majority of urinary amines, organic acids, and steroids in children. Together these results suggests that the majority of the urinary metabolites in children and blood metabolites in adults are determined by genetic and unique environmental influences and not by shared environmental influences.

It should be noted that the twin pairs in this study were selected for their concordance or discordance for childhood aggressive behavior, though we found no replicable associations between the urinary amines and organic acids with childhood aggression [33]. A limitation of the current study is the underrepresentation of DZ twin pairs. As a consequence, the power to detect significant dominant genetic and common environmental variance components was low [39,40]. It should also be noted that we often observed very large differences between the MZ and DZ twin correlations. This included instances where the MZ correlations were greater than four times the DZ correlations (4×*r_DZ_* < *r_MZ_*), which would result in negative values of the additive genetic variance in the ADE models [53]. Therefore, we chose to report the broad-sense heritability estimates for the metabolites with ADE models as opposed to the separate estimates for the additive and dominant genetic variance components. Based on the twin correlations, we can infer that dominant genetic effects likely contribute to the broad-sense heritability. This is in contrast to previous studies for serum metabolites in adults that suggested most genetic effects on metabolites were additive [8,9]. The pattern of very high MZ and very low DZ correlations could also indicate emergenesis (i.e., interactions between relatively infrequent alleles at different loci) [36,54], though it has been argued that duplicate gene interactions (i.e., complete dominance at both gene pairs, where the effect of one gene is hidden if the other is dominant) between moderately frequent alleles would also account for very low correlations between DZ twins [55].

In summary, this is the largest heritability study for urinary metabolites in children to date, benefitting from a large sample size (*N* = 1300), and the breath of coverage both in terms of the number of analyzed metabolites (86) and coverage across the metabolome by targeting amines, organic acids, and steroid hormones. We found substantial heritability for urinary metabolites in children, and genetic factors and not common environmental factors explain most of the familial resemblance in urinary metabolites. Age, but not sex, showed robust associations with urinary metabolites. These results are promising for future genome-wide association studies on urinary metabolites, which can elucidate the genetic architecture of the urinary metabolome in children.

## 4. Materials and Methods

### 4.1. Study Population and Procedures

#### 4.1.1. NTR-ACTION Cohort

Twin pairs from the Netherlands Twin Register [35] were invited for participation in the NTR-ACTION Biomarker Study (Aggression in Children: Unraveling gene-environment interplay to inform Treatment and InterventiON strategies) [31,32] based on their longitudinal concordance or discordance for aggressive behavior rated by either the mother (93%) or teacher (7%) (see [33]). Parents of participating twin pairs were asked to collect first-morning urine and buccal cell samples for their children using the standardized protocols as developed for the large-scale collection in the home situation (http://www.action-euproject.eu/content/data-protocols, accessed on 28 April 2022). Urine samples were stored in a home freezer at −18 °C, transported to the lab in a mobile freezer unit at −18 °C, and stored in the lab at −80 °C until further processing. Zygosity assessment in the twins was based on SNP arrays as described in Odintsova et al. [56].

First-morning urine was collected for 1362 twins, for whom metabolite profiling was performed on three metabolomics platforms targeting amines, steroids, and organic acids. We excluded twins if there was an insufficient amount of urine to analyze all three platforms (*N* = 2), if the collected urine was not the first-morning urine (e.g., parent-reported time of urine collection was after 12:00 in the afternoon; *N* = 13), if the time between collection and freezing of the urine sample was more than 2 h (*N* = 25), or if the twins were the second pair of multiples in the same family (*N* = 22). We performed the analyses in 1300 twins (Table 1). We give participant characteristics per aggression concordance group in Appendix A.

#### 4.1.2. LUMC-Curium Cohort

Children referred to the LUMC-Curium academic center for child and youth psychiatry and whose parents had consented to participation in the ongoing biobank protocol of LUMC-Curium were included in the ACTION Biomarker Study. Parents of participating children collected first-morning urine and buccal cell swabs with the protocols as developed for the NTR-ACTION cohort and all biological samples were analyzed simultaneously with the NTR samples. Information on psychiatric disorders in the LUMC-Curium sample is provided in Hagenbeek et al. (2020) [33]. In the LUMC-Curium cohort, we performed the analyses in 179 children (*M* age = 10.2, *SD* age = 1.8, age range: 6.3–13.4, 25.1% females, including 8 sibling pairs and 1 sibling trio), after exclusion of one child who also participated in NTR-ACTION, and exclusion of urine samples that were not first-morning (*N* = 3), or if the time between collection and freezing of the urine sample was more than 2 h (*N* = 3).

### 4.2. Creatinine Measurement

Creatinine was measured using a colorimetric assay kit according to manufacturer’s instructions (Cayman, Ann Harbor, MI, USA).

### 4.3. Metabolite Profiling

#### 4.3.1. Measurement Protocol

The Metabolomics Facility of the University of Leiden (Leiden, The Netherlands) assessed metabolites in urine on three platforms: an ultra-performance liquid chromatography mass spectrometry (UPLC-MS) platform targeting amines, a gas chromatography mass spectrometry (GC-MS) platform targeting organic acids, and a UPLC-MS platform targeting steroids. To ensure random distribution of aggression cases and controls across batches, we randomized NTR and LUMC-Curium ACTION subjects across batches. In each batch, we retained twins from the same pair on the same plate. Each batch included a calibration line, quality control (QC) samples (every 10 samples), sample replicates and blanks. The QC samples comprised pooled aliquots of all urine samples from all children. Using the pooled QC samples, the lab applied in-house developed algorithms to compensate for shifts in the sensitivity of the mass spectrometer across batches. Relative Standard Deviation (RSD) of the Quality Control samples (RSDqc) were used to evaluate the performance and reproducibility of individual metabolites. We report metabolites as ‘relative response ratios’ (target area/area of internal standard) after QC correction.

#### 4.3.2. LC-MS Amine Platform

Prior to chromatic separation, they added methanol to 5 μL of spiked (with internal standards, see Appendix A) urine for protein precipitation and centrifuged the supernatant. After sample evaporation (speedvac), samples were reconstituted in borate buffer (pH 8.5) and derivatized employing an AccQ-Tag derivatization strategy adapted from the protocol supplied by Waters (Waters, Etten-Leur, The Netherlands). An Agilent 1290 Infinity II LC system (1290 Multicolumn Thermostat and 1290 High Speed Pump, Agilent Technologies, Waldbronn, Germany) with an Accq-Tag Ultra column (Waters Chromatography B.V., Etten-Leur, The Netherlands) achieved chromatographic separation. The UPLC was coupled to electrospray ionization on a quadrupole-ion trap (SCIEX Qtrap 6500; Sciex, Framingham, MA, USA). Analytes were monitored in Multiple Reaction Monitoring (MRM) using nominal mass resolution and detected in the positive ion mode.

#### 4.3.3. GC-MS Organic Acid Platform

In order to extract the organic acids and remove urea, twice liquid-liquid extraction with ethyl acetate was applied to 50 μL of spiked (with internal standards: Succinic acid-2,2,3,3-d_4_, Fumaric acid-2,3-d_2_, and Citric acid-2,2,4,4-d_4_) urine. The two-step online derivatization procedure comprised oximation with methoxyamine hydrochloride (MeOX, 15 mg/mL in pyridine), followed by N-Methyl-N-(trimethylsilyl)- trifluoroacetamide (MSTFA) silylation. After derivatization, sample (1 μL) was injected into the GC-MS. With helium as a carrier gas (1.7 mL/min), chromatic separation was performed on a 25 m (HP-5MS UI) film thickness 30 × 0.25 m ID column. The mass spectrometer (MSD 5975C, Agilent Technologies, Waldbronn, Germany) was operated using a single quadrupole with electron impact ionization (70 eV) in SCAN mode (mass range *m*/*z* 50–500).

#### 4.3.4. LC-MS Steroid Platform

An LC-MS steroid platform was developed for the current study. Sample preparation comprised adding internal standards (Cortisone-2,2,4,6,6,9,12,12-d_8_, and Testosterone-16,16,17-d_3_) to 90 μL of urine and filtering the samples with a 0.2 μm PTFE membrane. Using an Acquity UPLC CSH C18 column, 130 Å, 1.7 µm, 2.1 mm × 100 mm, (Waters, Etten-Leur, The Netherlands), with a flow of 0.4 mL/min over a 15 min gradient, chromatographic separation was achieved by UPLC (Agilent 1290, San Jose, CA, USA). Samples were analyzed using a triple quadrupole mass spectrometer (Agilent 6460, San Jose, CA, USA) with electrospray ionization. By switching positive and negative ion mode, analytes were detected in MRM using nominal mass resolution.

#### 4.3.5. Metabolomics Data Preprocessing

We removed 4 amines with a missing rate of >10% and excluded metabolites with a relative standard deviation of the quality control samples (RSDqc) equal to or larger than 15% (excluded 6 amines, 1 organic acid, and 3 LC-MS steroids). For the remaining metabolites, we imputed metabolite measurements that fell below the limit of detection/quantification with half of the value of this limit, or when this limit was unknown with half of the lowest observed level for this metabolite. All metabolites were normalized by dividing by the sample creatinine levels and inverse normal rank transformed.

### 4.4. Statistical Analyses

#### 4.4.1. Associations with Sex and Age

To assess the influence of sex and age on urinary metabolites we ran mixed-effect models with sex and age as fixed effects and family and zygosity as random effects (to correct for family clustering in the data) (R v. 4.1.1, nlme v. 3.1–153) [57,58]. We expect that some metabolites have lower levels in younger children and will therefore not show sex differences at younger ages. Therefore, we performed a series of sensitivity and stratified analyses to elucidate the associations with sex and age further. We repeated the mixed-effect models with sex and age after dichotomizing age (age < 10 vs. age ≥ 10). We chose age 10 as cut-off for dichotomizing the age variable as 10 is the median age in the NTR and LUMC-Curium cohorts. Next, we performed stratified analyses. First, we investigated whether different metabolites show an association with age amongst males and females with mixed-effect models with age as fixed effect and family and zygosity as random effects. Again, these analyses were conducted for age as a continuous measure and after dichotomizing. Second, we employed mixed-effect models with sex as fixed effect and family and zygosity as random effects to investigate whether different metabolites show an association with sex amongst the younger (age < 10) and older (age ≥ 10) children. For each analysis we corrected for multiple testing separately with a False Discovery Rate (FDR) of 5% for 86 metabolites setting the significance threshold to *q* ≤ 0.05, i.e., 5% of the significant results will be false positives [59]. Thus, in the stratified analyses we correct separately in males and females or younger and older children. We replicated the analyses for sex and age for the urinary metabolites in 179 children of the LUMC-Curium cohort, by using mixed-effect models with family as random effects. We employed FDRs of 5% (*q* ≤ 0.05) for 79 (sex + continuous age), 76 (sex + dichotomized age), 73 (sex-stratified continuous age in males), 67 (sex-stratified continuous age in females), 63 (sex-stratified dichotomized aged in males), 51 (sex-stratified dichotomized aged in females), 11 (age-stratified sex in children age < 10), and 13 (age-stratified sex in children age ≥ 10) metabolites, respectively.

#### 4.4.2. Genetic Analyses

To quantify the contribution of genetic and environmental influences to complex traits, such as metabolites, the classical twin design (CTD) is often used [60]. The CTD hinges on comparing the resemblance between monozygotic (MZ) and dizygotic (DZ) twin pairs. MZ twin pairs derive from the same fertilized egg and share 100% of their genetic material. In contrast, DZ twin pairs derive from separately fertilized eggs and share on average 50% of their genetic material. Thus, by examining the difference between MZ and DZ covariances for a phenotype of interest, we can derive estimates for the proportions of variance attributable to genetic factors, i.e., heritability (h^2^), and to environmental factors. The CTD aims to decompose the phenotypic covariance between MZ and DZ twin pairs into additive genetic (A), non-additive or dominant genetic (D), shared or common environmental (C), and unshared or unique environmental factors (E) [61]. From models that estimate both additive and dominant genetic factors, we can obtain broad-sense h^2^ estimates, whereas models without dominant genetic factors allow us to estimate the narrow-sense h^2^ (or additive genetic).

We first fitted two group (MZ + DZ twin pairs) saturated models with sex and age as covariates and equal means and variances across birth order and zygosity by raw-data maximum likelihood (ML) in openMx (v. 2.19.8) [62] to obtain ML estimates of the twin correlations. We compared the saturated model with models that removed the age and/or sex covariates via likelihood ratio tests. We obtained the MZ and DZ twin correlations from the best-fitting model for each urinary metabolite (Appendix A). Next, we applied genetic structural equation modelling with raw-data ML estimation in openMx to estimate the additive genetic, dominant genetic or common environmental, and unique environmental variances for the urinary metabolites in the NTR-ACTION cohort based on the twin correlations (Figure 2, Appendix A). We implemented ADE models if the MZ correlations were greater than two times the DZ correlations (*r_MZ_* > 2×*r_DZ_*), and ACE models if this was not the case (*r_MZ_* < 2×*r_DZ_*). In our genetic models we did not constrain the variance components to take values between zero and one, thus, we allowed for negative variance estimates [53]. ADE or ACE models were estimated including sex and/or age as covariates depending on best-fitting saturated model (Appendix A). In the ADE model, we focus on the broad-sense h^2^, rather than reporting the separate estimates of the standardized variance of A and D. We did not test the significance of the D or C variance components, as the power to detect these variance components is low, given the relatively small number of complete DZ pairs (114) compared to the number of MZ pairs (645) (Table 1) [39,40].

## Figures and Tables

**Figure 1 metabolites-12-00474-f001:**
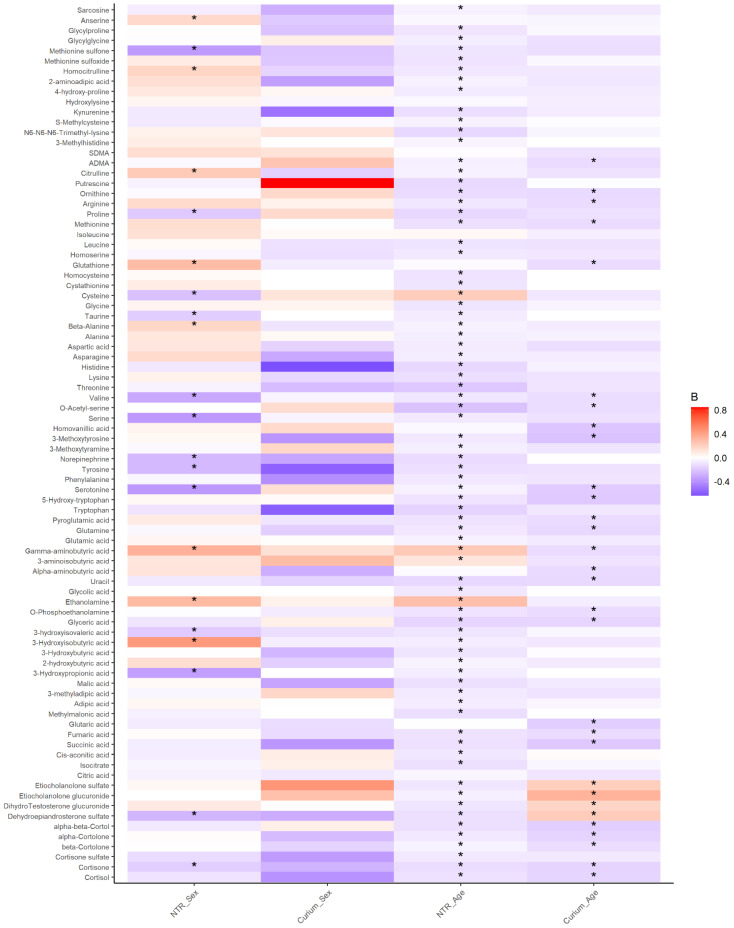
Associations of sex and age with metabolites in the NTR and LUMC-Curium ACTION cohorts. The vertical axis lists the metabolites, and the horizontal axis depicts the associations for sex or age in the discovery NTR-ACTION cohort (NTR) and the replication LUM-Curium ACTION cohort (Curium). All associations marked with a star (*) where significant after correcting for multiple testing (see Section 4). The color gradient indicates the strength of the associations (beta coefficient [B]), where dark blue indicates strong negative associations and dark red indicates strong positive associations. All association results from the discovery are reported in Appendix A and all association results from the replication are reported in Appendix A.

**Figure 2 metabolites-12-00474-f002:**
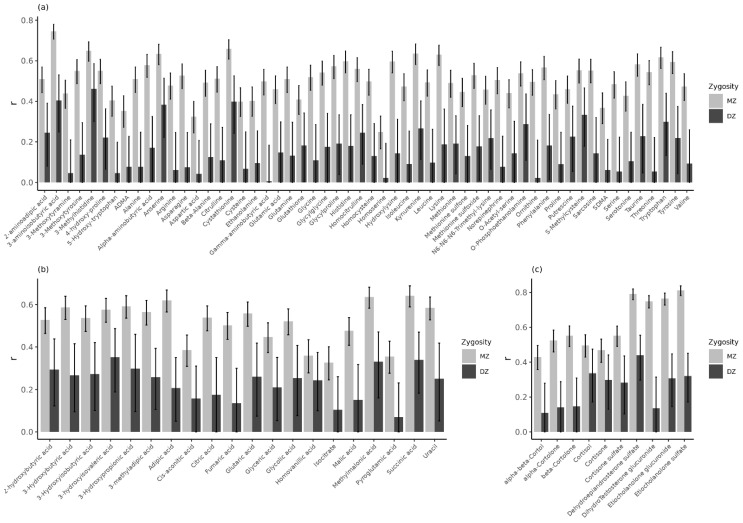
Monozygotic (MZ) and dizygotic (DZ) twin correlations of the (**a**) LC-MS amines, (**b**) GC-MS organic acids, and (**c**) LC-MS steroids corrected for sex and/or age in the NTR-ACTION cohort. See Appendix A for an overview of the correction for sex and/or age and for all correlations see Appendix A.

**Figure 3 metabolites-12-00474-f003:**
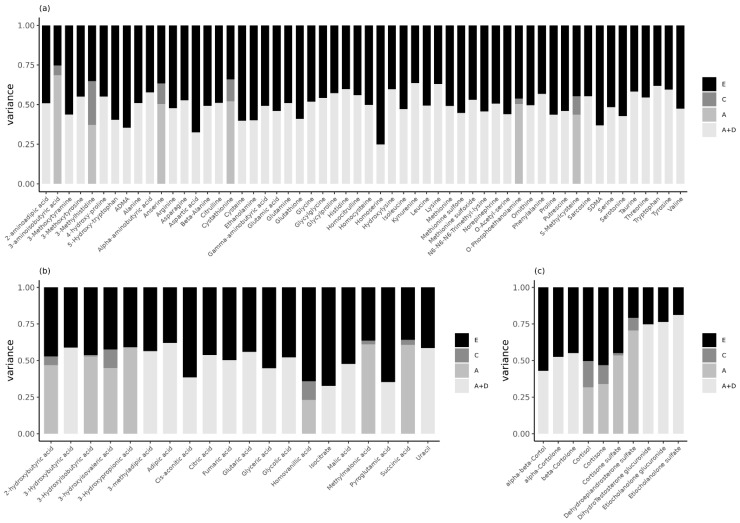
Estimates of the standardized variance components of the ADE or ACE models for the (**a**) LC-MS amines, (**b**) GC-MS organic acids, and (**c**) LC-MS steroids corrected for sex and/or age in the NTR-ACTION cohort. For the ADE models, we provide the estimates for the broad-sense heritability, i.e., the sum of the additive (A) and dominant (D) genetic variance components. See Appendix A for an overview of the correction for sex and/or age, and for all estimates and confidence intervals see Appendix A.

**Figure 4 metabolites-12-00474-f004:**
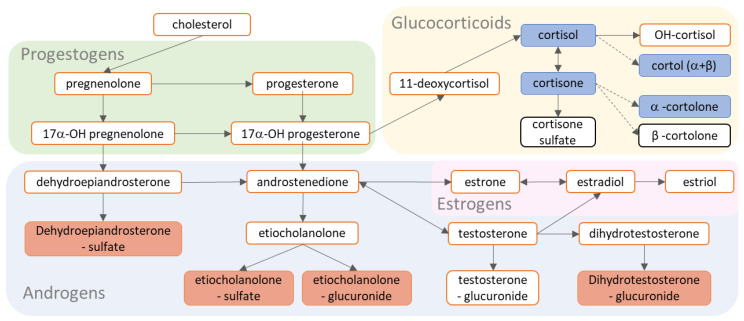
Metabolism of steroid hormones covered in this study. Metabolite boxes with red background indicate metabolites that were positively associated with age, a dark blue background indicates negative association with age. Metabolite boxes with white background in red frames were not covered in this study, whereas those in black frames were but did not show significant association with age in both cohorts. Arrows indicate direct reactions, whereas broken arrows indicate multi-step conversion. All association results for all models in the discovery are reported in Appendix A–S12 and all association results for all models in the replication are reported in Appendix A–S16.

**Table 1 metabolites-12-00474-t001:** Participant characteristics of the NTR and LUMC-Curium ACTION cohorts.

Cohort		*N* (*N* Pairs)	Mean (*SD*) [Range] Age	*N* (%) Females
NTR	Total	1300 (645)	9.6 (1.8) [5.7–12.9]	626 (48.2%)
	MZ	1068 (531)	9.6 (1.9) [6.0–12.9]	506 (47.4%)
	DZ	232 (114)	9.9 (1.6) [5.7–12.0]	120 (51.7%)
LUMC-Curium	Total	179	10.2 (1.8) [6.3–13.4]	45 (25.1%)

Notes: LC-MS, liquid chromatography mass spectrometry; GC-MS, gas chromatography mass spectrometry; NTR, Netherlands Twin Register; MZ, monozygotic twin pairs; DZ, dizygotic twin pairs.

## Data Availability

The standardized protocol for large scale collection of urine and buccal-cell samples in the home situation as developed for the ACTION Biomarker Study in children is available at http://www.action-euproject.eu/content/data-protocols (accessed on 28 April 2022). The data of the Netherlands Twin Register (NTR) ACTION Biomarker Study may be accessed, upon approval of the data access committee, through the NTR (https://tweelingenregister.vu.nl/information_for_researchers/working-with-ntr-data (accessed on 28 April 2022)).

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
