# Peer review of "Heritability of Urinary Amines, Organic Acids, and Steroid Hormones in Children"

_metabolites, 2022, doi:10.3390/metabo12060474_

Round 1

Reviewer 1 Report

I commend the authors for the wonderful work which will be of interest to the readers of this journal.

I have a few minor questions

1) Please provide some information on the Action biomarker study, while you have cited references, it will be informative to know the objective and logic behind the study.

2) You have reported that till date 3000 metabolites have been identified in urine, what was the reason for limiting this study to the 86 metabolites that you measured in this study.

3) Line 337 suggests SNP data was available for this study, wouldn't it be better to use SNPs for estimating heritability? and it would have also opened up interesting possibilities, such as performing  a metabolite wide association study to identify genomic loci associated with individual metabolites.

4) Will it be possible to deposit the LCMS, GC-MS raw spectra  generated in this study, after anonymizing individual identifiers in a public repository ?

Author Response

We are grateful to the reviewer for the compliment and for raising four points for improving our manuscript.

  1. Thank you for this question. The ACTION Biomarker Study was designed to investigate the etiology of childhood aggression and to assess the role of the genome, epigenome, and metabolome in explaining differences between children in aggressive behaviors [doi:10.1007/s00787-018-1169-1; doi:10.7363/040251; doi:10.3389/fpsyt.2020.00165]. The ACTION project included the largest genome-wide association study to date for childhood aggression (https://doi.org/10.1038/s41398-021-01480-x), a meta-analysis of DNA methylation signatures of aggression across the lifespan (https://doi.org/10.1038/s41380-020-00987-x), studies among academic experts' on their perspectives of prevalence, implementation, and utility of clinical guidelines for severe behavior problems in children (http://dx.doi.org/10.1016/j.eurpsy.2018.12.009) and documents the substantial comorbidity of aggression with other childhood disorders (https://doi.org/10.1007/s00787-018-1169-1 [http://www.action-euproject.eu/ComorbidityChildAggression] & https://doi.org/10.1371/journal.pone.0238667 [http://www.action-euproject.eu/TeacherRatingsChildAggression]).
    We added a brief statement regarding the ACTION Biomarker Study to the final paragraph of the Introduction (p. 2 lines 82-84.
  2. Thank you for pointing out that our reasoning for a targeted approach with a relatively limited number of metabolites was not clear. We have now clarified our choice for a targeted approach in the final paragraph of the Introduction on p. 2 lines 84-91: “To assess the metabolome, a targeted approach was chosen to cover metabolite classes previously associated with aggressive behavior [34], and urine samples were collected in twins from the Netherlands Twin Register [35] (NTR-ACTION cohort) and in an in-dependent clinical cohort (LUMC-Curium cohort). Specifically, urinary metabolomics data for 86 metabolites were generated on a liquid chromatography mass spectrometry (LC-MS) platform targeting amines, a gas chromatography (GC) MS platform targeting organic acids, and an LC-MS platform targeting steroid hormones.”
    In addition, we would like to note that the actual coverage of our three metabolomics platforms is higher than the 86 metabolites discussed in the manuscript. As described in the Methods p. 10-11 lines 370-426, these 86 metabolites are those metabolites that survived quality control.
  3. The reviewer is correct. For the participants in the ACTION Biomarker Study, we have genotype data available as measured with the Affymetrix Axiom or Illumina GSA genotyping arrays (for details see our preprint that comprises the genetic, epigenetic, and metabolomics data from the ACTION Biomarker Study [https://doi.org/10.1101/2021.09.13.21263063]).
    It certainly is an important next step to estimate the SNP-based heritability for our urinary metabolites and to compare our estimates for overlapping metabolites with those as obtained in Calvo-Serra et al. (2021; doi:10.1093/hmg/ddaa257). However, our cohort does not reach the N=3000 guidelines to reliably estimate SNP-based heritability through GCTA approaches (https://doi.org/10.1371/journal.pgen.1004269).
    As stated in the Discussion on p. 10 lines 328-330, we also fully concur with the reviewer that performing genome-wide association studies for our urinary metabolites is of interest, however, this is beyond the scope of the current manuscript.
  4. It is standard policy within the Netherlands Twin Register, that all data, including the metabolomics data of the NTR-ACTION Biomarker Study, may be accessed upon reasonable request and after approval of the data access committee. (https://tweelingenregister.vu.nl/information_for_researchers/working-with-ntr-data)

Reviewer 2 Report

The review by Hagenbeek and coauthors investigates the role of genetic and environmental factors in urinary metabolomics data in children. In particular, the associations of 86 urinary metabolites with sex and age were investigated in two different children cohorts including monozygotic (MZ) and dizygotic (DZ) twins, thus allowing to assess also the reliability of the performed measures. Then, the heritability of the metabolites was investigated by estimating the correlations in MZ and DZ twin pairs and by applying genetic structural equation modeling of MZ and DZ data. The authors found that genetic factors and not common environmental factors explain most of the familial resemblance in urinary metabolites. Furthermore, age, but not sex, showed the most robust associations with urinary metabolites.

In my opinion, this work helps to elucidate the genetic architecture of the urinary metabolome in children, thus it can be considered for publication. I suggest only minor revisions:

Rephrase the abstract in order to make it more fluent to read, I would suggest removing some of the mean values (and the relative intervals).

Please specify from the beginning which are the two cohorts (NTR and LUMC-Curium ACTION) and their relative acronyms.

Author Response

We appreciate the endorsement for publication of the reviewer and thank the reviewer for raising two points for improving our manuscript.

  1. We thank the reviewer for encouraging us to enhance the readability of the abstract. To conform with the reviewer’s suggestion to remove some of the mean values (and ranges) we have removed the mean monozygotic and dizygotic twin correlations and replaced the mean estimates (and ranges) with an interpretation of the twin correlations in terms of the best-fitting genetic model (i.e., whether they indicated genetic models including non-additive (dominant) genetic or shared (common) environmental factors would best fit). To better distinguish between those metabolites with non-additive genetic or shared environmental factors, we also rewrote how we presented the mean broad- and narrow-sense heritability estimates.
  2. Thank you for alerting us to the fact that we did not provide a clear description, including the appropriate acronyms, for our cohorts. On advice of reviewer #1, we added a brief statement regarding the ACTION Biomarker Study to the final paragraph of the Introduction (p. 2 lines 82-91) where we now define the two cohorts and give their acronyms: “The ACTION Biomarker Study was designed to investigate the etiology of childhood aggression and to assess the role of the genome, epigenome, and metabolome in explaining differences between children in aggressive behaviors [31–33]. To assess the metabolome, a targeted approach was chosen to cover metabolite classes previously associated with aggressive behavior [34], and urine samples were collected in twins from the Netherlands Twin Register [35] (NTR-ACTION cohort) and in an independent clinical cohort (LUMC-Curium cohort). Specifically, urinary metabolomics data for 86 metabolites were generated on a liquid chromatography mass spectrometry (LC-MS) platform targeting amines, a gas chromatography (GC) MS platform targeting organic acids, and an LC-MS platform targeting steroid hormones.”
    To increase consistency across the final paragraph of the Introduction, we have also adjusted the sentences on p. 2 lines 91-94 (“Here, we first investigate the effects of sex and age on these 86 urinary metabolites in the NTR-ACTION cohort comprising 1068 monozygotic (MZ) twins (531 pairs, M age = 9.6, SD age = 1.9, 47.4% females) and 232 dizygotic (DZ) twins (114 pairs, M age = 9.9, SD age = 1.6, 51.7% females, Table 1).”) and p. 2 lines 94-96 (“We undertook to replicate the sex and ages associations in the LUMC-Curium cohort of 179 children (M age = 10.2, SD age = 1.8, age range: 6.3-13.4, 25.1% females, Table 1).”).